# Using Laser-Doppler Flowmetry to Evaluate the Therapeutic Response in Dentin Hypersensitivity

**DOI:** 10.3390/ijerph17238787

**Published:** 2020-11-26

**Authors:** Mariana Miron, Diana Lungeanu, Edmond Ciora, Emilia Ogodescu, Carmen Todea

**Affiliations:** 1Department of Oral Rehabilitation and Dental Emergencies, Faculty of Dentistry, “Victor Babes” University of Medicine and Pharmacy, P-ta Eftimie Murgu 2, 300041 Timisoara, Romania; miron.mariana@umft.ro (M.M.); todea.darinca@umft.ro (C.T.); 2Center for Modeling Biological Systems and Data Analysis, Department of Functional Sciences, “Victor Babes” University of Medicine and Pharmacy, P-ta Eftimie Murgu 2, 300041 Timisoara, Romania; 3Department of Pediatric Dentistry, Faculty of Dentistry, “Victor Babes” University of Medicine and Pharmacy, P-ta Eftimie Murgu 2, 300041 Timisoara, Romania; ogodescu.emilia@umft.ro

**Keywords:** dentin hypersensitivity, microcirculation, laser Doppler flowmetry, therapeutic research, outcome assessment

## Abstract

Dentin hypersensitivity (DH) is a common medical condition with underreported prevalence and it is difficult to quantify. This study aimed to investigate whether assessing dental pulp vascular micro-dynamics by laser-Doppler flowmetry (LDF) would be functional for therapeutic evaluation, in contrast to a verbal rating scale (VRS). A split-mouth single-blind randomized study was conducted on seven patients and a total of 36 teeth. Two DH therapeutic methods were employed: (i) fluoride gel; (ii) Nd:YAG radiation combined with fluoride gel. For each tooth, five consecutive LDF determinations of pulp blood flow were made (before and immediately after desensitizing treatment, then after 24 h, 7 days, and 1 month), and the VRS was applied each time. Spearman’s correlation was applied for concurrent validation. Two-way (treatment and patient) repeated measures ANOVA full factorial was applied, followed by Tukey’s post-hoc comparisons and Pillai’s trace multivariate statistic. While VRS scores had moderate reliability, LDF could objectively estimate treatment effects. Based on partial eta-squared values, treatment and patient characteristics were estimated to explain about 84% and 50% of the variability, respectively. In conclusion, LDF is an objective technique that can quantitatively assess DH evolution, and it is effective in reliably monitoring oral health therapeutic interventions.

## 1. Introduction

According to international studies [1,2,3,4], dentin hypersensitivity (DH) is a complex condition with multi-factorial etiology and high prevalence in all age groups, which may severely affect quality of life [5].

DH is rapidly triggered when dentinal tubules are exposed to mechanical, thermal, and/or chemical stimuli temporarily present in the oral cavity. It manifests as a sharp, short-lived pain. The etiology is complex and the pathogenesis is unclear, although there are several theories that try to explain the mechanism. Brannstrom’s hydrodynamic theory [6,7] is widely accepted and states that when the dentin is exposed, the stimuli determine fluid movements in the open dentinal tubules, generating negative or positive pressures on the nerve endings of the plexus around the odontoblasts. This leads to nerve fibers’ mechanical deformation, Na^+^ ions channels’ broadening, and fibers’ depolarization, thereby causing pain [7]. Oral parafunctional habits, such as bruxism or mouth breathing, are also associated with DH [4,5].

Current methods to evaluate cervical DH or a desensitizing agent’s effectiveness assess the intensity of the dental pulp response to various stimuli (e.g., low temperatures according to hydrodynamic theory). One-dimensional pain scales such as the Numerical Rating Scale (NRS), Verbal Rating Scale (VRS), or Visual Analogue Scale (VAS) are usually employed to assess pain intensity (PI) [8,9]. On the other hand, laser-Doppler flowmetry (LDF) has been reported as an objective, non-invasive, and real-time method for assessing the dental pulp status [10,11,12,13,14,15].

DH treatment is complex and widely varied [16,17,18,19,20,21,22,23,24,25,26,27,28,29,30,31,32,33], with tailored strategies according to particular etiological factors. In practice, we aim to prevent direct contact between external stimuli and tubular fluid, mainly by closing the dentinal tubules. When dentin hypersensitivity is confirmed, in the absence of tooth structure lesions and any contributory medical status, the usual therapeutic approach is as follows: (a) reducing risk factors by educating the patient to avoid acids in the diet and perform correct oral hygiene; (b) correcting possibly unhealthy habits and/or parafunctions; (c) instructing the patient on proper brushing technique; (d) recommending the use of desensitizing agents at home; (e) professional application of local desensitizing methods, by the dentist’s office specialist. Among the DH therapeutic approaches, laser therapy has been reported as successful [21,22,23,24,25,26,27,28,29,34,35,36], although there is no standard protocol yet.

In clinical practice, evaluating treatment efficiency could be an issue, as current one-dimensional pain scales make it difficult to compare treatments and therapeutic strategies. Therefore, building evidence towards standardized practical protocols for assessing and comparing therapeutic interventions in DH is of clinical relevance.

The primary aim of this study was to investigate whether using LDF could be a functional method to assess dental pulp vascular micro-dynamics in the DH therapeutic response. This research focused on whether LDF recordings were consistent with subjective pain scales, rather than being spurious measurements. As pain scales and LDF are not equivalent in their sensitivity, this was not a study on the medical equivalency of the two instruments. The secondary aim was to compare two treatment approaches and measure the treatment effect size for each one. On the basis of its corresponding statistical hypotheses, the required sample size was determined.

## 2. Materials and Methods

### 2.1. Therapeutic Methods Employed

Two treatments were applied: (i) desensitizing gel with amorphous calcium phosphate (ACP), a casein derivative proven to be effective in treating DH (2% fluoride gel, Relief ACP Dental Oral Care Gel, 2.4 g syringe, Philips Oral Healthcare, Stamford, USA, as shown in Figure 1) [30,31]; (ii) ACP gel followed by Nd:YAG (neodymium-doped yttrium aluminum garnet; Nd: Y_3_Al_5_O_12_) laser radiation therapy applied directly through the gel, with a Fidelis plus II 300 FOTONA equipment (FOTONA, Ljubljana, Slovenia). The laser therapy parameters are presented in Table 1. Figure 2 shows the working technique during the treatment sessions.

### 2.2. Evaluating the Therapeutic Response

Two evaluation instruments were employed to assess and quantify the treatment effects: (i) a VRS for the intensity of experienced pain triggered by applying compressed air for up to 10 s, on a scale from 0 (absence of pain) to 4 (unbearable pain) [8,9,37,38]; (ii) LDF for recording the vascular micro-dynamics at the dental pulp level [13,14,15].

To assess the therapeutic response with LDF, the equipment comprised the following: a MoorLab laser-Doppler device for general medical use (Laser Doppler MoorLab instrument VMS-LDF2, Moor Instruments Ltd., Axminster, UK); straight optic probe VP3, with a length of 10 mm, built to be used on the oral mucosa/teeth. The laser-Doppler signal acquisition technique was performed according to our previous studies [39,40].

In order to stabilize the laser probe in the tooth’s cervical third, a double silicone impression was taken using Kit Optosil Comfort Putty and Xantopren Comfort Light, Haereus (Heraeus Kulzer, GmbH Leipziger Straße 2, 63450 Hanau, Germany). This is a silicone-based condensation curing material that takes impressions with high dimensional stability in moist dental environments, thus allowing precise reproduction detail (Figure 3). The impression was further used as an LDF probe holder for acquiring laser-Doppler signals.

For LDF recordings, a heat-free light-cured liquid dam (LC Block-Out Resin, Ultradent, Products GmbH, Am Westhover Berg 30, 51149 Cologne, Germany) was applied around every tooth, radius 3–4 mm (Figure 4a). This liquid dam helped to reduce secondary signals generated by the gingival micro-circulation, as these could spoil the actual pulp blood flow signal. Then, the tooth’s surface was cleansed and the LDF probe holder was positioned as in Figure 4b.

For each measurement, the pulp blood flow signal was recorded for 1.5 min. Vascular micro-dynamics at the dental pulp level were measured as pulsatory signals, as shown in Figure 5. The flow is related to the product of the average speed and concentration of mobile red blood cells in the tissue sample volume. The digital signal’s mean values and standard deviations were recorded for further analysis.

### 2.3. Study Design

A split-mouth single-blind randomized study was conducted using a mid-sagittal plane between the central incisor teeth [41]. The study was designed as a mixed factorial setup, combining the between-group treatment factor with the within-subject variable of repeated measures in time.

The research focused on the change in the mean blood flow value, so the necessary sample size was decided based on this expected effect size and a balanced two-way ANOVA model. For the experimental model with two treatments and at least five patient groups, the necessary size resulted in three teeth in each group, leading to the required total of at least 30 teeth in the sample (alpha = 0.05; beta = 0.2; Cohen’s d effect size was equal to 0.8 for both treatment and within-subject factors).

The study was performed at the Municipal Hospital of Timisoara, Romania, affiliated with the “Victor Babes” University of Medicine and Pharmacy in Timisoara. The study design and protocol were approved by the Ethics Committee of the Municipal Hospital (Approval No I-26730/12.11.2020).

### 2.4. Sample Description and Study Protocol

Seven patients (aged between 22 and 35) were enrolled in the study from September 2018 to December 2019, each with six or four single-rooted anterior teeth in two symmetrical quadrants, resulting in a total of 14 quadrants and 36 teeth (larger than the required sample size of 30 teeth, for a level of confidence 1-alpha = 0.95 and a statistical power 1-beta = 0.8, as determined for this study design). The inclusion criteria were as follows: frontal teeth hypersensitivity (maxillary or mandibular), with DH pain between 2 and 3 on the VRS; selected teeth had to be vital, decay free and without restorations; patients agreed to follow the brushing technique and instructions on proper hygiene throughout the study. The exclusion criteria were as follows: professional desensitizing treatment or any use of desensitizing toothpaste in the last year; etiological conditions that determine or predispose to dentinal hypersensitivity, such as bruxism or other parafunctions; acid-rich diet or exposure to acids; chronic use of anti-inflammatory medication, pain killers, and/or psychotropic drugs; pregnancy; allergic manifestations or idiosyncrasies to various products; food-related syndromes associated with regurgitation; periodontal surgery or orthodontic treatment in the last three months; teeth with conservative treatments in the last six months, or teeth having large obturations that extended into the area to be tested; pillar teeth of fixed or mobile dentures.

Each participant signed an informed consent. For each patient, an even number of frontal teeth were examined, equally distributed on the left and right quadrants, as shown in Figure 6. For each participant, symmetric teeth were randomly assigned into one of the two treatment groups.

The patients were instructed on maintaining good oral hygiene and a healthy diet, and they were asked not to use dental floss, anti-inflammatories, or desensitizing medication throughout the study. In addition, they were instructed to avoid brushing at least one hour before each assessment session.

For each tooth included in the study, five paired assessments were made, comprising VRS scores and LDF recordings: at baseline (i.e., before the desensitizing treatment), immediately after treatment, after 24 h, after 7 days, and after 1 month.

### 2.5. Data Analysis

#### 2.5.1. Analysis of Verbal Rating Scale Scores

VRS reliability was evaluated using intraclass correlation coefficients (ICCs), for the two arms of the study at post-treatment times and overall. Descriptive statistics included the mean (M), standard deviation (SD), standard error of the mean (SEM), and mean detectable change (MDC). The Wilcoxon non-parametric statistical test was applied to compare the matched scores in the two study groups.

#### 2.5.2. Analysis of Laser Doppler Flowmetry Measurements

Descriptive statistics of the blood flow included the mean (M) and standard deviation (SD) for each combination of categorical variables, at the five measurement times. Two-way ANOVA was applied to compare the mean blood flow values. The variability of the laser-Doppler flowmetry recordings was described by the median and inter-quartile range for the standard deviations of 1.5 min recordings for each tooth, at the five measurement times. For each measurement time, the distribution of flow variability across the treatments was compared with the Kolmogorov-Smirnov nonparametric test.

The actual effect of each treatment on each tooth at each post-treatment assessment was quantified by the relative difference in mean blood flow compared to baseline. Repeated measures ANOVA full factorial was applied across the two factors (patient and treatment), followed by post-hoc comparisons between the treatments according to the Tukey procedure, with confidence intervals for the estimated treatment effect.

#### 2.5.3. Analysis of Concurrent Validity

Spearman correlation was used to assess the validity of dentin sensitivity determined in this study. At the five measurement times, the correlation coefficients were calculated overall, and for each treatment group.

All reported probability values were two-tailed. A 0.05 level of significance was employed, and highly significant values were also marked. Data were analyzed with the statistical software IBM SPSS v.20.0 (Armonk, New York, USA) and R v.3.2.3 software packages [42], including the package “pwr2”.

## 3. Results

Appendix A shows the actual Laser-Doppler flowmetry measurements for the 36 teeth.

Table 2 presents the VRS descriptive statistics for the two study groups. Taking advantage of the split-mouth design, the Wilcoxon non-parametric test was applied to compare the related scores for the corresponding teeth in the two study groups.

Table 3 shows the ICCs values for post-treatment VRS scores (for each study group and overall), proving moderate reliability of the actual scores, although with high statistical significance.

Table 4 presents the variability in flowmetry values in the laser-Doppler recordings. There was no statistical difference in the distribution of standard deviations in the two treatment groups.

The mean values of the recordings were further analyzed. Table 5 shows the Spearman correlation coefficients between VRS scores and LDF mean values, proving the concurrent validity of dentin sensitivity measurements in this study and the subsequent analysis.

Table 6 presents the descriptive statistics for the LDF measurements in the two treatment groups, at each assessment time. The two-way ANOVA showed a statistically significant difference between teeth activity as recorded by the laser-Doppler investigation at each post-treatment measurement moment. At baseline, there were no differences between the two treatment groups. For all four post-treatment measurements, there was a statistically significant difference between the two treatment effects, although the immediate effect was in the opposite direction compared to the long-/medium-term effect (i.e., 30-day effect). There was no significant interaction between patient and treatment in the 30-day observation period: from a statistical point of view, based on the raw values, the treatment effect was independent of the patient’s characteristics.

To assess and compare the treatment effect at subsequent measurement times, irrespective of patient characteristics, the relative difference in mean blood flow for each tooth was calculated at each post-treatment assessing time compared to the initial value (Table 7). There was no significant interaction between the patient and treatment effect in the medium-/long-term measurements.

The repeated measures ANOVA with Pillai’s trace multivariate statistic was applied to further assess the time effect (Table 8). The effect of time itself, and of both treatment and patient proved to be statistically significant. In addition, the partial eta-squared values showed about 84% and 50% of the effect variability was explained by the treatment and patient’s characteristics, respectively (as ratios to the respective group differences plus associated error variance).

Figure 7 shows the treatments’ marginal effects over the four post-treatment measurement times in this study. One can see the reversed differences in medium-/long-term effect (7 days and 30 days) compared to the immediate after treatment effect. Table 9 synthesizes the estimated effect size with 95% confidence level.

## 4. Discussion

Dentin sensitivity measurements in this study had acceptable concurrent validity, with statistically significant correlation between the VRS scores and LDF recordings, and an overall range between 0.6 and 0.79 for the Spearman correlation coefficients. This confirms a good concordance between the objective LDF measurements and the clinical symptoms, with subjective reaction to pain confirming earlier reports on using LDF to assess pulp status [11,12,13,14,15]. Not surprisingly, in this study the correlation was lower in case of Nd:YAG laser treatment upon immediate assessment: the VRS scores might have been impaired, while the LDF measurements reflected the microcirculation storm related to laser exposure.

In spite of their inherently subjective nature, the VRS scores proved to be moderately reliable, confirming the reports regarding this pain scale being dependable on a wide range of medical conditions and preferred by cognitively intact young adults [37,38,43,44]. On the other hand, with rather large MDC values for small samples (as in this study), VRS could only illustrate one-dimensional differences between treatment groups, signaling the occasional blurred boundaries and triggering hypotheses, as in case of Nd:YAG laser group immediately after exposure. MDC aims to define the change that can be genuinely attributed to the intervention and not impaired by measuring errors. For the two treatment groups in this study, the 30-day difference in VRS scores was 0.61 points, while MDC was over 2.8. Moreover, the ICC in the Nd:YAG laser group was considerably lower compared to the control gel group. This reduced reliability was generated by VRS scores taken immediately after treatment.

The shortcomings of pain scales are counterbalanced by the very nature of continuous numerical values in LDF recordings, which allow multi-dimensional analysis. They can also capture and quantify interaction and time effects. The LDF measurements and the full-factorial ANOVA model in this study quantified the between-treatment differences, as well as the after-treatment immediate and medium-/long-term effect and differences. In addition, LDF could pinpoint the significant interaction between treatment effect and patients’ characteristics, and it quantified their actual contribution to the overall effect of therapeutic intervention. Furthermore, the treatment effect could be quantitatively estimated. Therefore, LDF can objectively assess the effect of an oral healthcare intervention, and it can avoid pain threshold alterations due to sensitization or habituation effects in the case of repeated painful stimuli, which are unavoidable and uncontrollable when employing sensibility testing to assess the therapeutic response [10,11,12,13,14,15].

Regarding the secondary aim of this study, the evaluation of pulp vascular micro-dynamics tested at the four post-treatment times highlighted the following aspects: immediately after the combined gel and Nd:YAG laser therapy, the LDF recordings showed a significantly intensified effect in the case of the exposed teeth, as compared to the control gel group. Moreover, the pulsatile flow characteristics of the laser-Doppler signal recorded immediately after the combined therapy was significantly altered, probably due to laser radiation. All patients reported a light but painful sensitivity during laser radiation and shortly after. This immediate effect was also reflected in the VRS scores for the combined therapy group. On the other hand, the medium-/long-term effect was significantly better, with substantially decreased DH in the group of combined gel and Nd:YAG laser therapy, compared to the control gel group. This significant treatment effect was assessed by both the VRS scores and the LDF measurements, and it was quantified by LDF. The post-treatment transition time interval was significantly dependent on the patients’ characteristics, from one to several days, but all seven patients were in a steady-state condition after one week (as the relative difference in LDF measurements confirmed). The results of this study support the findings already reported in the literature [21,26,36].

### 4.1. Study Limitations

This study had certain limitations due to the controlled experimental conditions: (a) it was conducted by highly trained medical professionals; (b) all the patients complied with strict requirements regarding their home hygiene and eating regime. These factors might raise issues concerning reproducibility in ordinary practical settings. To be used in a clinical setting, the LDF procedure should be standardized. In addition, the necessary sample size of this study was determined based on the LDF-detectable effect, which led to high MDC values in VRS scores and, therefore, hampered their sensitivity. However, the aim of the study was to prove that LDF would be sensitive and accurate in measuring the effect size in small samples and even for individual care.

### 4.2. Future Perspectives

Further investigations with larger samples would help in building stronger evidence towards using LDF for gauging therapeutic effects in DH treatment. Furthermore, LDF should be evaluated with a wider variety of therapeutic approaches in ordinary dental practice.

## 5. Conclusions

LDF reliably recorded the changes occurring in dental pulp microcirculation at all assessment times of the study, even when the patient did not identify any change from a clinical point of view through VRS.

This was a proof-of-concept study, demonstrating that LDF would be an objective instrument to evaluate and quantify the effects of DH treatment. LDF can capture the contribution of many concurrent factors and their interactions, in contrast to the subjective one-dimensional pain scales used in current practice. LDF would also be effective in reliably monitoring the oral health status; therefore, it is of clinical relevance. As a sensitive and reliable instrument, it could be employed in screening and prevention programs.

## Figures and Tables

**Figure 1 ijerph-17-08787-f001:**
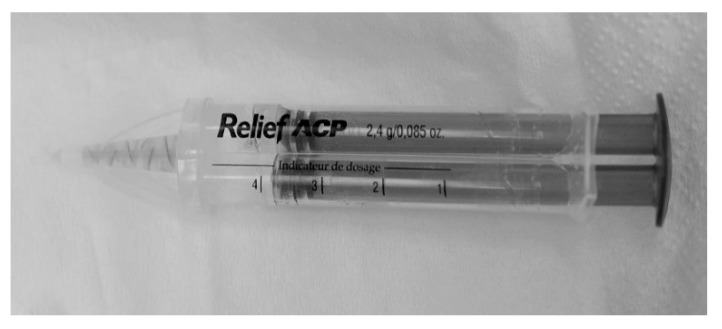
Amorphous calcium phosphate (ACP) desensitizing gel was employed in this study.

**Figure 2 ijerph-17-08787-f002:**
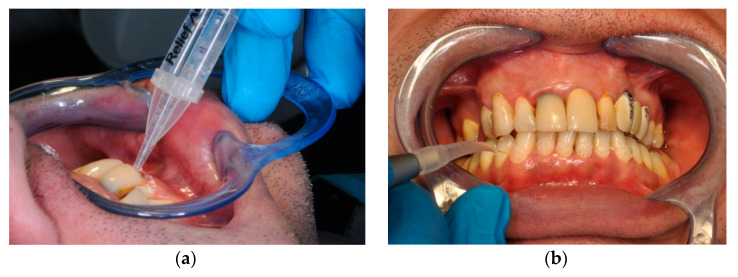
Treatment application: (**a**) desensitizing gel in the cervical vestibular area; (**b**) Nd:YAG laser radiation, directly through the gel, on a tooth in the fourth quadrant.

**Figure 3 ijerph-17-08787-f003:**
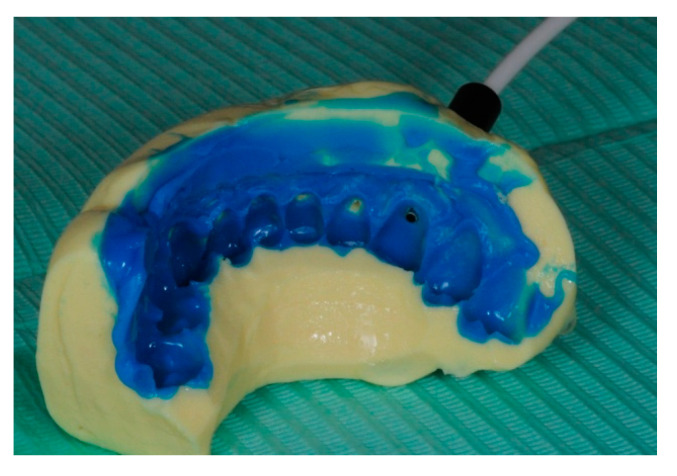
Position of the optic probe in the impression.

**Figure 4 ijerph-17-08787-f004:**
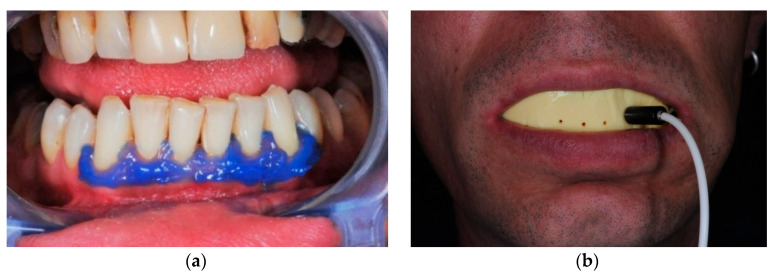
Laser-Doppler flowmetry (LDF) recording process: (**a**) periodontal liquid dam as gingival barrier applied around every tooth; (**b**) position of the impression with the LDF probe.

**Figure 5 ijerph-17-08787-f005:**
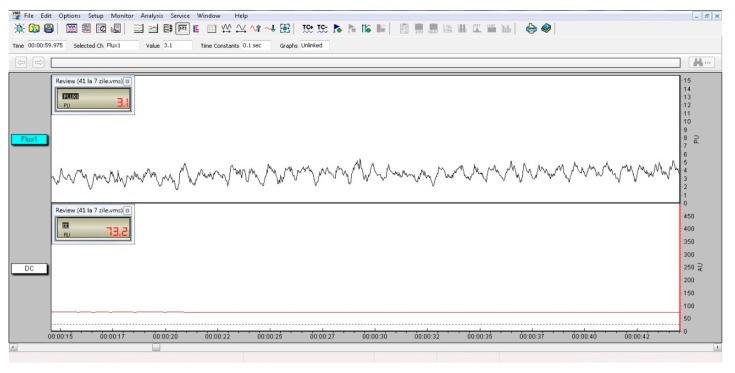
Laser Doppler pulp level signal for tooth 41, seven days after the treatment. The pulsatory signal is shown in the upper window.

**Figure 6 ijerph-17-08787-f006:**
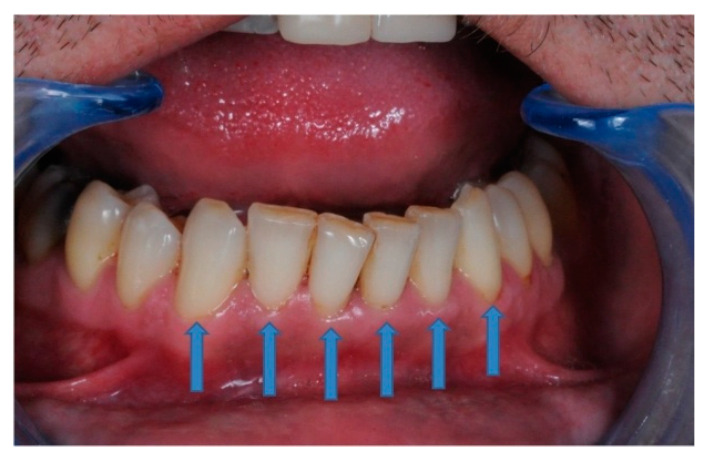
Example of symmetric frontal teeth allocation in the study (43, 42, 41, 31, 32, and 33 in this image).

**Figure 7 ijerph-17-08787-f007:**
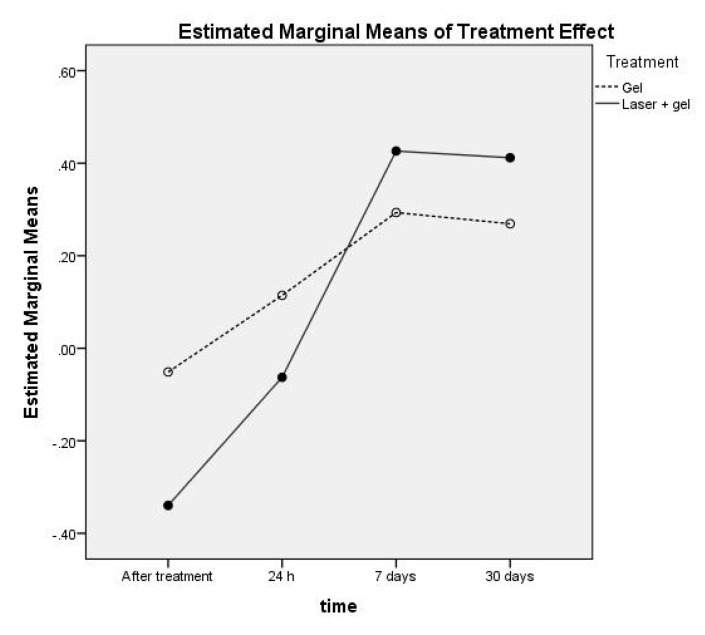
Estimated effect size for the two treatments in the very short and medium term. For the numerical values on the vertical axis, the null stands to the left of the decimal point.

**Table 1 ijerph-17-08787-t001:** Nd:YAG (neodymium-doped yttrium aluminum garnet; Nd: Y_3_Al_5_O_12_) laser parameters for therapeutic intervention in the study.

Nd:YAG Laser Parameters	
λ wavelength	1064 nm
Optic probe	300 µm
Operation mode	VSP
Power	1.00 W
Frequency	10 Hz
Exposure time/tooth	15 sec.
Number of applications/sessions	4
Number of sessions	1

VSP stands for Very Short Pulse.

**Table 2 ijerph-17-08787-t002:** Verbal rating scale (VRS) descriptive statistics for the two study groups.

Time	Gel (*n* = 18)		Laser + Gel (*n* = 18)		*p* ^(a)^
	M ± SD	SEM (MDC)	M ± SD	SEM (MDC)	
Initial	2.50 ± 0.514	0.121 (2.893)	2.56 ± 0.511	0.121 (2.893)	0.655
After treatment	2.56 ± 0.511	0.121 (2.893)	2.89 ± 0.323	0.076 (2.848)	0.034 *
24 h	2.28 ± 0.575	0.135 (2.907)	2.39 ± 0.502	0.118 (2.890)	0.157
7 days	2.06 ± 0.539	0.127 (2.899)	1.56 ± 0.511	0.121 (2.893)	0.003 **
30 days	1.94 ± 0.416	0.098 (2.870)	1.33 ± 0.485	0.114 (2.886)	0.001 **

^(a)^ Wilcoxon nonparametric statistical test; M—mean, MDC—mean detectable change, SD—standard deviation, SEM—standard error of the mean; * *p* < 0.05; ** *p* < 0.01.

**Table 3 ijerph-17-08787-t003:** Intraclass correlation coefficients (ICCs) for VRS reliability data.

ICCs	Overall (*n* = 36)	Gel (*n* = 18)	Laser + Gel (*n* = 18)
Estimator value	0.424 **	0.605 **	0.460 **
95% Confidence interval	(0.255; 0.603)	(0.382; 0.801)	(0.223; 0.705)

Statistical significance: ** *p* < 0.001.

**Table 4 ijerph-17-08787-t004:** Variability of flowmetry values in the laser-Doppler 1.5 min recordings: standard deviation of values recorded for individual teeth.

Time	Gel ^(a)^ (*n* = 18)	Laser + Gel ^(a)^ (*n* = 18)	*p* ^(b)^
Initial	1.35 (0.9–1.5)	1.2 (1.0–1.9)	0.766
After treatment	1.25 (1.1–1.6)	1.85 (1.3–2.3)	0.057
24 h	1.0 (0.8–1.3)	1.5 (1.1–1.6)	0.057
7 days	0.9 (0.8–1.1)	0.8 (0.7–1.1)	0.491
30 days	1.0 (0.8–1.1)	0.8 (0.6–1.4)	0.057

^(a)^ median (Inter-Quartile Range), with Tukey’s hinges; ^(b)^ Kolmogorov-Smirnov statistical test.

**Table 5 ijerph-17-08787-t005:** Concurrent validity of the dentin sensitivity measurements: Spearman correlation coefficients between VRS scores and Laser Doppler flowmetry recordings.

Time	Overall (*n* = 36)	Gel (*n* = 18)	Laser + Gel (*n* = 18)
Initial	0.739 **	0.696 **	0.787 **
After treatment	0.651 **	0.701 **	0.443
24 h	0.788 **	0.805 **	0.793 **
7 days	0.676 **	0.631 **	0.690 **
30 days	0.599 **	0.275	0.580 *

Statistical significance: * *p* < 0.05; ** *p* < 0.01.

**Table 6 ijerph-17-08787-t006:** Descriptive statistics of flowmetry values in the laser-Doppler 1.5 min recordings for the seven patients enrolled in this split-mouth study.

Time		Gel ^(a)^	Laser + Gel ^(a)^	Two-Way ANOVA
				Flowmetry by Patient, Treatment
Initial	Patient 1 (*n* = 3 + 3)	6.50 ± 0.87	6.27 ± 0.9	Model: *p* < 0.001 ** PatientID: *p* < 0.001 ** Treatment: *p* = 0.602 Two-way Interaction PatientID * Treat: *p* = 0.375
	Patient 2 (*n* = 3 + 3)	5.03 ± 1.88	5.63 ± 1.32	
	Patient 3 (*n* = 2 + 2)	8.35 ± 1.48	5.55 ± 1.06	
	Patient 4 (*n* = 2 + 2)	6.35 ± 0.7	4.95 ± 0.49	
	Patient 5 (*n* = 3 + 3)	10.20 ± 2.38	11.00 ± 1.32	
	Patient 6 (*n* = 3 + 3)	15.83 ± 0.7	16.27 ± 1.01	
	Patient 7 (*n* = 2 + 2)	7.30 ± 1.7	8.20 ± 1.41	
After treatment	Patient 1 (*n* = 3 + 3)	6.97 ± 0.86	9.13± 0.50	Model: *p* < 0.001 ** PatientID: *p* < 0.001 ** Treatment: *p* < 0.001 ** Two-way Interaction PatientID * Treat: *p* = 0.891
	Patient 2 (*n* = 3 + 3)	5.97 ± 2.12	7.03 ± 0.68	
	Patient 3 (*n* = 2 + 2)	8.3 ± 0.85	10.55 ±1.77	
	Patient 4 (*n* = 2 + 2)	7.95 ± 0.07	8.55 ± 1.77	
	Patient 5 (*n* = 3 + 3)	9.60 ± 1.35	11.90 ± 1.57	
	Patient 6 (*n* = 3 + 3)	15.33 ± 0.51	17.50 ± 0.66	
	Patient 7 (*n* = 2 + 2)	6.60 ± 1.13	8.40 ± 1.41	
24 h	Patient 1 (*n* = 3 + 3)	5.23 ± 0.45	6.13 ± 1.83	Model: *p* < 0.001 ** PatientID: *p* < 0.001 ** Treatment: *p* = 0.019 * Two-way Interaction PatientID * Treat: *p* = 0.787
	Patient 2 (*n* = 3 + 3)	6.20 ± 2.29	5.67 ± 0.85	
	Patient 3 (*n* = 2 + 2)	5.40 ± 1.27	7.45 ± 1.77	
	Patient 4 (*n* = 2 + 2)	5.40 ± 0.14	7.45 ± 1.77	
	Patient 5 (*n* = 3 + 3)	8.37 ± 2.71	10.23 ± 1.10	
	Patient 6 (*n* = 3 + 3)	14.13 ± 0.40	15.70 ±0.70	
	Patient 7 (*n* = 2 + 2)	5.75 ± 0.07	6.85 ± 1.91	
7 days	Patient 1 (*n* = 3 + 3)	4.60 ± 0.44	2.83 ± 0.51	Model: *p* < 0.001 ** PatientID: *p* < 0.001 ** Treatment: *p* = 0.007 ** Two-way Interaction PatientID * Treat: *p* = 0.461
	Patient 2 (*n* = 3 + 3)	4.67 ± 1.50	3.93 ± 0.76	
	Patient 3 (*n* = 2 + 2)	4.05 ± 0.35	2.45 ± 0.35	
	Patient 4 (*n* = 2 + 2)	4.90 ± 0.14	3.30 ± 0.14	
	Patient 5 (*n* = 3 + 3)	6.43 ± 1.81	6.90 ± 1.66	
	Patient 6 (*n* = 3 + 3)	10.77 ± 1.44	8.57 ± 1.39	
	Patient 7 (*n* = 2 + 2)	4.45 ± 0.49	4.05 ± 0.49	
30 days	Patient 1 (*n* = 3 + 3)	4.60 ± 0.35	2.43 ± 0.15	Model: *p* < 0.001 ** PatientID: *p* < 0.001 ** Treatment: *p* = 0.002 ** Two-way Interaction PatientID * Treat: *p* = 0.725
	Patient 2 (*n* = 3 + 3)	4.10 ± 1.15	3.70 ± 0.17	
	Patient 3 (*n* = 2 + 2)	5.30 ± 0.42	3.30 ± 0.14	
	Patient 4 (*n* = 2 + 2)	4.45 ± 0.21	3.30 ± 0.14	
	Patient 5 (*n* = 3 + 3)	7.47 ± 2.64	7.10 ± 1.27	
	Patient 6 (*n* = 3 + 3)	11.00 ± 0.20	9.60 ± 1.40	
	Patient 7 (*n* = 2 + 2)	5.70 ± 0.71	4.45 ± 0.07	

^(a)^ M ± SD; * *p* < 0.05; ** *p* < 0.01.

**Table 7 ijerph-17-08787-t007:** Treatment effect size as the relative difference between flowmetry values in the post-treatment laser-Doppler 1.5 min recordings for each tooth compared to baseline.

Time		Gel Effect ^(a)^	Laser + Gel Effect ^(a)^	Two-Way ANOVA
				Effect by Patient, Treatment
After treatment	Patient 1 (*n* = 3 + 3)	−0.07 ± 0.06	−0.47 ± 0.17	Model: *p* < 0.001 ** PatientID: *p* < 0.001 ** Treatment: *p* < 0.001 ** Two-way Interaction PatientID * Treat: *p* = 0.004 **
	Patient 2 (*n* = 3 + 3)	−0.19 ± 0.05	−0.28 ± 0.23	
	Patient 3 (*n* = 2 + 2)	−0.02 ± 0.28	−0.91 ± 0.05	
	Patient 4 (*n* = 2 + 2)	−0.25 ± 0.3	−0.72 ± 0.19	
	Patient 5 (*n* = 3 + 3)	0.05 ± 0.09	−0.09 ± 0.13	
	Patient 6 (*n* = 3 + 3)	0.03 ± 0.02	−0.08 ± 0.04	
	Patient 7 (*n* = 2 + 2)	0.09 ± 0.06	−0.06 ± 0.35	
24 h	Patient 1 (*n* = 3 + 3)	0.18 ± 0.16	0.01 ± 0.30	Model: *p* = 0.003 ** PatientID: *p* = 0.042 * Treatment: *p* = 0.001 ** Two-way Interaction PatientID * Treat: *p* = 0.008 **
	Patient 2 (*n* = 3 + 3)	−0.24 ± 0.18	−0.02 ± 0.12	
	Patient 3 (*n* = 2 + 2)	0.33 ± 0.27	−0.34 ± 0.06	
	Patient 4 (*n* = 2 + 2)	0.15 ± 0	−0.49 ± 0.21	
	Patient 5 (*n* = 3 + 3)	0.19 ± 0.08	0.06 ± 0.10	
	Patient 6 (*n* = 3 + 3)	0.11 ± 0.06	0.03 ± 0.05	
	Patient 7 (*n* = 2 + 2)	0.19 ± 0.38	0.13 ± 0.38	
7 days	Patient 1 (*n* = 3 + 3)	0.28 ± 0.15	0.54 ± 0.06	Model: *p* = 0.038 * PatientID: *p* = 0.018 * Treatment: *p* = 0.017 * Two-way Interaction PatientID * Treat: *p* = 0.778
	Patient 2 (*n* = 3 + 3)	0.06 ± 0.04	0.26 ± 0.31	
	Patient 3 (*n* = 2 + 2)	0.50 ± 0.13	0.54 ± 0.15	
	Patient 4 (*n* = 2 + 2)	0.23 ± 0.03	0.33 ± 0.10	
	Patient 5 (*n* = 3 + 3)	0.37 ± 0.07	0.38 ± 0.10	
	Patient 6 (*n* = 3 + 3)	0.32 ± 0.12	0.47 ± 0.12	
	Patient 7 (*n* = 2 + 2)	0.37 ± 0.22	0.49 ± 0.15	
30 days	Patient 1 (*n* = 3 + 3)	0.29 ± 0.08	0.48 ± 0.05	Model: *p* = 0.003 ** PatientID: *p* = 0.04 * Treatment: *p* < 0.001 ** Two-way Interaction PatientID * Treat: *p* = 0.158
	Patient 2 (*n* = 3 + 3)	0.17 ± 0.10	0.32 ± 0.16	
	Patient 3 (*n* = 2 + 2)	0.36 ± 0.06	0.40 ± 0.09	
	Patient 4 (*n* = 2 + 2)	0.30 ± 0.03	0.33 ± 0.04	
	Patient 5 (*n* = 3 + 3)	0.28 ± 0.10	0.36 ± 0.06	
	Patient 6 (*n* = 3 + 3)	0.30 ± 0.04	0.41 ± 0.12	
	Patient 7 (*n* = 2 + 2)	0.21 ± 0.09	0.45 ± 0.10	

^(a)^ M ± SD; * *p* < 0.05; ** *p* < 0.01.

**Table 8 ijerph-17-08787-t008:** Repeated measures ANOVA for the treatment and patient effects over time (multivariate full factorial model).

Effect	Pillai’s Trace Value	F (df1, df2)	*p*	Partial Eta-Squared
Time	0.972	227.895 (3, 20)	<0.001 **	0.972
Time * Treatment	0.839	34.803 (3, 20)	<0.001 **	0.839
Time * PatientID	1.504	3.685 (18, 66)	<0.001 **	0.501
Time * Treatment * PatientID	1.172	2.350 (18, 66)	0.006 **	0.391

** high statistical significance (*p* < 0.001).

**Table 9 ijerph-17-08787-t009:** Estimated treatment effect as the relative difference between the post-treatment flowmetry values and the initial recorded values.

Treatment	Time	M	SEM	95% Confidence Interval	
				Lower Bound	Upper Bound
Gel	After treatment	−0.053	0.035	−0.126	0.021
	24 h	0.130	0.042	0.043	0.217
	7 days	0.304	0.035	0.231	0.376
	30 days	0.272	0.022	0.226	0.318
Laser + gel	After treatment	−0.371	0.035	−0.444	−0.298
	24 h	−0.087	0.042	−0.174	−0.001
	7 days	0.430	0.035	0.358	0.503
	30 days	0.409	0.022	0.363	0.455

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
