# Peer review of "Using Laser-Doppler Flowmetry to Evaluate the Therapeutic Response in Dentin Hypersensitivity"

_ijerph, 2020, doi:10.3390/ijerph17238787_

Round 1
Reviewer 1 Report
This manuscript untitled “Using Laser-Doppler Flowmetry as an Evaluation Method of Therapeutic Response in Dentin 3 Hypersensitivity”. Aim of this paper is quite interesting but is not new. Generally, there are grammatical errors in this manuscript. It is recommended that it would be revised again by the English scientific writer.
Introduction section must be rewritten because the presence of loose sentences does not help the reader, they must have a sequence. And you should use the paragraphs for different subjects to guide the reader.
Collectively, despite the changes, this manuscript still needs Major reviews to publish in IJERPH at this point. For these, there are numerous issues in the present manuscript that need to be addressed before publication:
Statement of Clinical Relevance
- What is the importance of this review for the clinical? You do not think this study is included to the others already done? Which results are comparable?
- What this study has new?
Introduction
- Is very small, and sometimes confusing.
- What is the novelty of this paper? Please clarify in the appropriate section.
- What was your hypothesis null hypothesis?
Materials and Methods
- Please include a statement in the Material and Methods section that the study has been approved by the institutional ethics committee and provide the number of the process.
- How was the sample calculated? Did authors perform a power analysis to evaluate if this sample size was appropriate?
- When mentioning materials or devices: for some of them you don't mention the manufacturer at all, for some you mention only the manufacturer, for some the manufacturer and city, for some you mention the manufacturer and city/ country.
- Improve the quality of all images, has little resolution.
- Figure 6 - Can you add a radiograph of these teeth?
Discussion
- Please, identified what was the principal limitations of this study? And also, future perspectives.
References
- Check reference’s format in the manuscript, and in the references. The titles of references have different format and identification of the name are also in different formats
Reviewer 2 Report
The topic is interesting in the conservative dentistry field because it deals with one of the problems (dentinal sensitivity) that the dentist is commonly forced to face despite having few means with results that are not always predictable.
The experimentation was conducted with correct methodology and statistical analysis, even if, in my opinion, the low number of patients included limits the results, perhaps it would have been better to wait for a greater number of cases.
However, given the absence of a protocol in the literature on the procedures for the use of pulp vascular micro-dynamics by laser-Doppler flowmetry (LDF) in the field of dentinal hypersensitivity, I believe that the article is worthy of publication.
Only a few aspects in my opinion and should be improved.
Introduction
What are the parafunctions associated with hypersensitivity?
Materials and methods
Has the trial been screened by an ethics committee (indicate protocol number)?
Have patients been evaluated for parafunctions?
Reason for the choice of desensitizing 2% fluoride gel Relief ACP 2.4g / 0.085oz as desensitizer?
Results
Could the instructions given to the patient on home hygiene and eating habits have influenced the results in the 2 groups (gel, gel + laser)? Without these instructions, the results would have been different?
Is it possible to improve the quality of clinical images?
How could the limitations of this study possibly have affected the results?
Round 2
Reviewer 1 Report
This research is under the scope of this journal. The topic is relevant to readers. And this research deals with potentially significant knowledge to the field and an open new way for future studies. Aim of this paper is quite interesting.
Authors performed the asked revisions in a proper form and the quality of the manuscript improved compared with the first version.The authors implemented all the requested changes. Thank you. Good work!
Reviewer 2 Report
the authors answered all the doubts inherent in the manuscript and made the necessary changes
Best regards